# Accuracy of Assessing Weight Status in Adults by Structured Observation

Tânia Jorge [1], Sofia Sousa [2,3,4], Isabel do Carmo [1], Nuno Lunet [3,4,5] and Patrícia Padrão [2,3,4,*]

[1] Faculdade de Medicina, Universidade de Lisboa, Avenida Professor Egas Moniz, 1649-028 Lisboa, Portugal; taniajsjorge@gmail.com (T.J.); isabel.carmo72@gmail.com (I.d.C.)

[2] Faculdade de Ciências da Nutrição e Alimentação, Universidade do Porto, Rua do Campo Alegre 823, 4150-180 Porto, Portugal; sofia.sousa@ispup.up.pt

[3] EPIUnit—Instituto de Saúde Pública, Universidade do Porto, Rua das Taipas 135, 4050-600 Porto, Portugal; nlunet@med.up.pt

[4] Laboratório para a Investigação Integrativa e Translacional em Saúde Populacional (ITR), Rua das Taipas 135, 4050-600 Porto, Portugal

[5] Departamento de Ciências da Saúde Pública e Forenses e Educação Médica, Faculdade de Medicina, Universidade do Porto Alameda Professor Hernâni Monteiro, 4200-319 Porto, Portugal

[*] Correspondence: patriciapadrao@fcna.up.pt; Tel.: +351-225074320; Fax: +351-225074329

**Abstract:** The assessment of weight status is important in many epidemiological studies, but its direct measurement is not always possible. Self-reported weight and height are often used, although previous research reported low accuracy. This study aimed to test the ability of trained observers to accurately estimate weight status in adults using structured observation. A cross-sectional study was conducted. For each participant, height and weight were estimated in categories, and weight status was recorded using Stunkard's body figures, by two trained observers. Height and weight were also measured, using standardized procedures. Subjects were classified according to World Health Organization body mass index (BMI) cut-offs from objective measurements and from the BMI assigned to each body figure. Sensitivity, specificity, and likelihood ratios were calculated to assess the accuracy of estimating weight status by observation. Kappa was used to test inter-observer reliability. A total of 127 participants were assessed, 70 women and 57 men, aged between 19 and 89 years (mean ± standard deviation: 50.3 ± 16.3 years). Most participants were overweight or obese (64.3% women; 78.9% men). The sensitivity and specificity of overweight/obesity status identification were 72.8% and 78.4%, respectively. Observers' gender, participants' gender, and participants' age were significantly associated with the estimation of overweight/obesity. The agreement between observers was moderate for BMI estimates (κ = 0.52) but substantial when distinguishing normal weight from overweight/obesity (κ = 0.67). Trained observers were able to distinguish normal weight from overweight/obesity with high sensitivity and specificity, and substantial interrater reliability. This innovative methodology showed potential for improvement through enhanced training techniques. The use of structured observation may be a useful and accurate alternative to self-reported weight status assessment, whenever anthropometric measurement is not achievable.

**Keywords:** weight; body mass index; anthropometric measures; body size figures; overweight and obesity

## 1. Introduction

Weight status is of interest in epidemiological studies both in estimating prevalence and its trends as well as in studies of disease prevention, risk assessment, co-morbidities, mortality, and the economic burden of the overweight and obesity epidemic. In large population studies, data on weight and height are often collected by self-reporting and then used to calculate body mass index (BMI) as one of the most popular measures to categorize participants as underweight, normal weight, overweight, or obese [1–3]. However,

previous studies have shown that self-reported weight and height are often inaccurate, with individuals underestimating their weight and overestimating their height, resulting in an underestimation of BMI [4–14].

Silhouette-based matching tests have been used to assess body image self-perception [15], since the BMI corresponding to the chosen silhouette has shown a high correlation with measured BMI in adults [16]. Specifically, assigning self-reported weight status in adults by the selection of the silhouette from the Stunkard Figure Rating Scale [17] has been reported to have a good correlation with measured BMI [18]. The validation of the Stunkard scale as an instrument to assess nutritional status was confirmed by Sorensen et al. [19]. There is, however, a lack of data on the validity of the weight status estimation performed by observers trained in using this type of silhouette. Studying the accuracy of estimating weight status by structured observation would be an important step in validating this new methodology, which could be used as a simple and quick way to collect weight status data as an alternative to self-reported values whenever it is not possible to perform anthropometric measures.

For the present investigation, it was hypothesized that estimated measures by observers trained in using body image scales might be used in assessing the weight status of adult individuals when it is not possible to perform anthropometric measures. As such, this study aimed to test the ability of trained observers to accurately classify adult individuals by structured observation regarding weight status, and to assess the concordance between observers in estimating the weight, height, and BMI categories of adult individuals.

## 2. Materials and Methods

The data used in the present study were obtained from a cross-sectional study conducted from May to June 2018 with a convenience sample of adults. Trained observers relied on direct observation to classify participants regarding categories of weight, height, and weight status [17].

### 2.1. Participant Selection and Recruitment

Participants were recruited after being admitted for blood collection in a public laboratory in Leiria (Portugal). All individuals able to stand up to obtain subjective and objective measures of weight and height were considered eligible for the present study. Those with clinical conditions that could interfere with weight and height measurements, such as edema, amputations, and orthopedic problems, as well as pregnant women, were excluded. While in the waiting room, each eligible individual was invited to join the study after being informed about its objectives and the procedures involved.

### 2.2. Recruitment and Training of the Observers

In order to recruit observers, an email was sent to the directors of all undergraduate courses at the School of Health Sciences of the Polytechnic of Leiria, asking for the dissemination of the study through their students, who were invited to collaborate as observers. A total of six students were interested in collaborating and were, thus, recruited. All observers were finishing their undergraduate degrees in Dietetics and Nutrition, except one, who was in the course of Physiotherapy. Three students in Dietetics and Nutrition were also healthcare workers with a previous degree and professional experience in another health field.

Training aimed at providing the observers with theoretical knowledge and practical skills for estimating weight status using a body image scale, as well as standardizing procedures. After explaining the study, its objectives, and methodology, the Stunkard Figure Rating Scale [17] was presented, including (1) its rationale and what it consists of; (2) the silhouettes that compose it; and (3) the BMI corresponding to each figure. Then, practical exercises were performed, in which the figures were randomly projected on a white board (three times each) and classified by the observers regarding the correspondent BMI, at the end of which the observers were given feedback regarding their evaluations.

### 2.3. Procedures for Data Collection

Each participant was asked to stay in front of a white wall, between one and two minutes, standing still and facing the observers. During this time period, the two observers positioned themselves in the frontal plane towards the participant at a distance of approximately three meters. According to their trained perception, each observer independently estimated (i.e., recorded individually and without exchanging information with their partner) the participants' weight and height categories (detailed in Section 2.3.1), as well as their weight status, by selecting one of the nine Stunkard's Figures.

After the observation, the participant was informed that this step of assessment had been completed, and they were accompanied to a separate room, where he/she was objectively evaluated by a third trained researcher, who performed direct anthropometric measurements using standard procedures [20,21] (detailed in Section 2.3.2), also collecting data on gender and age. There was no communication between these three researchers throughout the evaluations.

### 2.3.1. Measures Estimated by Observation

Height and weight were estimated in categories, as follows:

- Height: (1) $\leq$144 cm, (2) 145–154 cm, (3) 155–164 cm, (4) 165–174 cm, (5) 175–184 cm, (6) $\geq$185 cm;
- Weight: (1) $\leq$44 kg, (2) 45–54 kg, (3) 55–64 kg, (4) 65–74 kg, (5) 75–84 kg, (6) 85–94 kg, (7) $\geq$95 kg.

Due to the low percentage of participants in some categories, for data analysis, the variables were recoded into new categories, as follows:

- Height: (1) $\leq$154 cm, (2) 155–164, (3) $\geq$165 cm;
- Weight: (1) $\leq$54 kg, (2) 55–74 kg, (3) $\geq$75 kg.

BMI category estimates were given by the number of the figures of Stunkard's scale selected by the observer (from 1 to 9), each one identified with its corresponding BMI according to Bulik et al. [18], for females and males, respectively: Figure 1—18.3/19.8 kg/m$^2$; Figure 2—19.3/21.1 kg/m$^2$; Figure 3—20.9/22.2 kg/m$^2$; Figure 4—23.1/23.6 kg/m$^2$; Figure 5—26.2/25.8 kg/m$^2$; Figure 6—29.9/28.1 kg/m$^2$; Figure 7—34.3/31.5 kg/m$^2$; Figure 8—38.6/35.2 kg/m$^2$; Figure 9—45.4/41.5 kg/m$^2$. These were then classified according to the World Health Organization (WHO) cut-offs [18] into four classes: (1) underweight (BMI < 18.5 kg/m$^2$); (2) normal weight (BMI 18.5–24.9 kg/m$^2$); (3) overweight (BMI 25.0–29.9 kg/m$^2$); and (4) obese (BMI $\geq$ 30.0 kg/m$^2$).

### 2.3.2. Anthropometric Measures

Anthropometric measurements were carried out in accordance with the Portuguese Guideline "Procedimentos Antropométricos na Pessoa Adulta" [Anthropometric Procedures in the Adult Person] issued by the Directorate-General of Health [20] and the "International Standards for Anthropometric Assessment" [21]. Height was measured to the nearest 0.1 cm, using a SECA$^\circledR$ Portable Stadiometer HR001. Weight was measured to the nearest 100 g using a digital scale (TANITA$^\circledR$ TBF- 300A). BMI was calculated, and subjects were classified according to the WHO cut-offs [22].

### 2.4. Statistical Analyses

After confirmation of the normality of the data, the mean and standard deviation (SD) were used to describe the mean age and measured height, weight, and BMI of the 127 participants. These continuous variables were compared using the t-test for independent samples. Real age, height, weight, and BMI were also categorized and summarized as counts and percentages, in order to give a more detailed overview of the distributions of the participants. These categorical variables were compared using the Chi-square test.

Considering the 254 observations, Spearman's rank correlation was used to assess the associations between measured and estimated weight and height.

The sensitivity, specificity, and likelihood ratios (LR) of the 254 observations were calculated in order to assess the accuracy of estimating obesity and overweight/obesity by trained observers (overall, among male observers, and among female observers). Chi-square and Fisher's exact tests, as applicable, were performed to assess the association between correct identification of obese and non-obese individuals, and between overweight/obese and non-overweight/obese subjects according to the gender of the observer, the gender of the participant, and the age of the participant.

The inter-rater concordance of the 127 paired observations was quantified using the Kappa statistic, for estimated height, weight, Stunkard figures, and BMI categories.

Statistical analysis was conducted using IBM SPSS Statistics® version 28.0.0.0 Subscription for Macintosh Operating System and STATA® version 15.1 for Windows®. The level of significance was set at 0.05.

## 3. Results

### 3.1. Demographic and Anthropometric Measurements of Observers and Participants

Among the observers, four were women and two were men, aged from 23 to 41 years old. The mean ± SD weight and height of the observers were 66.0 ± 9.4 kg and 169.2 ± 9.2 cm, respectively. BMI ranged from 20.7 to 25.5 kg/m$^2$ (mean ± SD: 22.9 ± 1.7 kg/m$^2$).

Demographic and anthropometric data on participants are shown in Table 1. The sample included 127 participants, 70 women and 57 men, aged between 19 and 89 years (mean ± SD: 50.3 ± 16.3 years). Most participants were overweight or obese (64.3% women and 78.9% men).

**Table 1.** Age and anthropometric measures of the participants (n = 127).

| | Total (n = 127) | Women (n = 70) | Men (n = 57) | *p*-Value |
|---|---|---|---|---|
| Age (years), mean ± SD | 50.3 ± 16.3 | 47.9 ± 1.7 | 53.4 ± 2.5 | 0.058 |
| Age categories, n (%) | | | | |
| 18–34 | 24 (18.9) | 15 (21.4) | 9 (15.8) | |
| 35–54 | 53 (41.7) | 33 (47.1) | 20 (35.1) | 0.127 |
| ≥55 | 50 (39.4) | 22 (31.4) | 28 (49.1) | |
| Height (cm), mean ± SD | 164.9 ± 9.6 | 159.7 ± 0.8 | 171.5 ± 1.1 | <0.001 [a] |
| Height categories, n (%) | | | | |
| ≤154 | 14 (11.0) | 13 (18.6) | 1 (1.8) | |
| 155–164 | 52 (40.9) | 42 (60.0) | 10 (17.5) | <0.001 [b] |
| ≥165 | 61 (48.0) | 15 (21.4) | 46 (80.7) | |
| Weight (kg), mean ± SD | 73.1 ± 12.9 | 68.3 ± 1.4 | 79.1 ± 1.6 | <0.001 [a] |
| Weight categories, n (%) | | | | |
| ≤ 54 | 11 (8.7) | 10 (14.3) | 1 (1.8) | |
| 55–74 | 63 (49.6) | 41 (58.6) | 22 (38.6) | <0.001 [b] |
| ≥75 | 53 (41.7) | 19 (27.1) | 34 (59.6) | |
| BMI (kg/m$^2$), mean ± SD | 26.9 ± 4.1 | 26.8 ± 0.5 | 26.9 ± 0.5 | 0.900 |
| BMI categories, n (%) | | | | |
| <18.5 | 5 (3.9) | 4 (5.7) | 1 (1.8) | |
| 18.5–24.9 | 32 (25.2) | 21 (30.0) | 11 (19.3) | |
| 25.0–29.9 | 61 (48.0) | 24 (34.3) | 37 (64.9) | 0.006 [b] |
| ≥30 | 29 (22.8) | 21 (30.0) | 8 (14.0) | |

BMI, Body mass index; SD, standard deviation. [a] Statistically significant differences according to the *t*-test for independent samples with a significance level of 0.05. [b] Statistically significant differences according to the Chi-square test with a significance level of 0.05.

### 3.2. Associations between Measured and Estimated Weight and Height

Figure 1 shows the distributions of measured weight and height, according to the respective categories of estimation (n = 254 observations). A strong positive rank correlation

was observed between the anthropometric measures and the corresponding estimations, both for weight ($\rho = 0.74$, $p < 0.001$) and height ($\rho = 0.78$, $p < 0.001$).

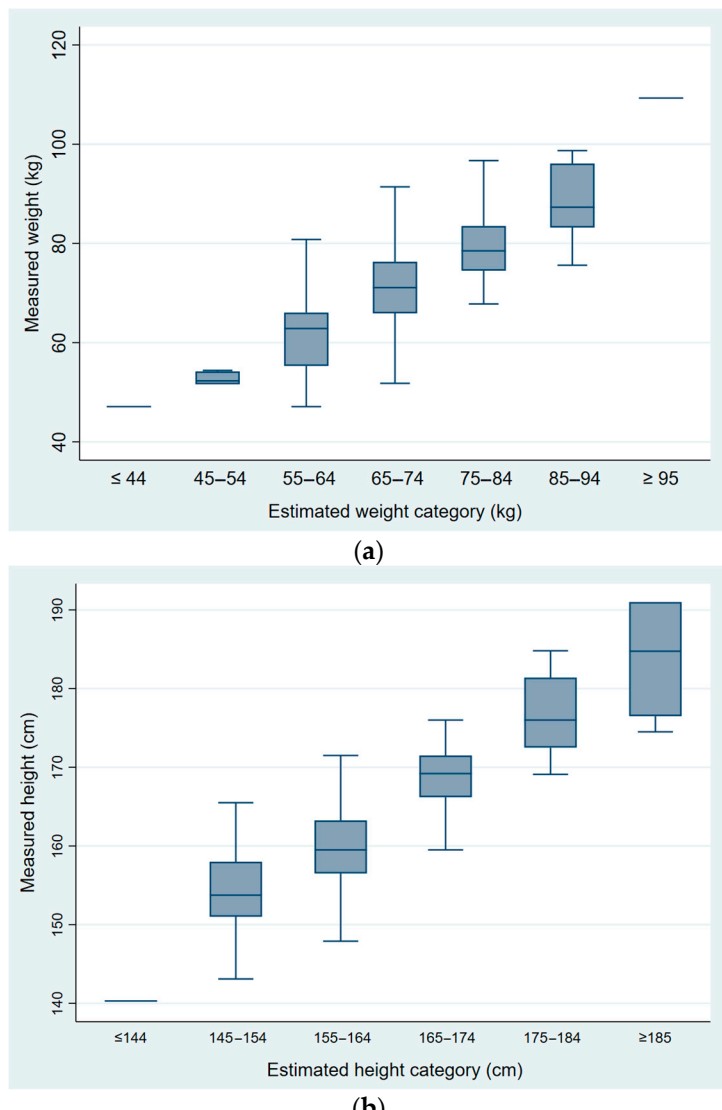

**Figure 1.** Distributions of measured weight (**a**) and height (**b**), according to the respective categories of estimation (n = 254 observations).

### 3.3. Validity of Estimating BMI by Trained Observers

As shown in Table 2, sensitivity was higher for estimating overweight/obesity status than for identifying obesity status alone (72.8% vs. 41.4%). Specificity was higher for estimating obesity than for identifying overweight/obese subjects (96.4% vs. 78.4%).

**Table 2.** Sensitivity, specificity, and likelihood ratios of estimated measures: obesity, overweight/obesity (n = 254 observations).

| Participants' Characteristics | Sensitivity | Specificity | Likelihood Ratio | |
|---|---|---|---|---|
| | | | Positive (LR+) | Negative (LR−) |
| Weight status | | | | |
| Obesity | 41.4% | 96.4% | 11.5 | 0.61 |
| Overweight/obesity | 72.8% | 78.4% | 3.4 | 0.35 |

For obesity, correct classification of obese subjects was more than 11 times more likely than misclassification of the non-obese (LR+ = 11.5), while the probability of misclassification of the obese was nearly 40% lower than the probability of correct classification of the non-obese (LR− = 0.61). For the combined status of overweight and obesity, the probability of correct classification of the overweight/obese subjects was more than three-fold higher than that of misclassification of the non-overweight/obese (LR+ = 3.4), while the probability of misclassification of the overweight/obese was nearly one-third the probability of correct classification of the non-overweight/obese (LR− = 0.35).

In Table 3, gender of observer, gender of participant, and age of participant were shown to be associated with the estimation of obesity and overweight. Women observers classified obesity with higher sensitivity than male observers (56.8% vs. 14.3%, *p* = 0.002). When considering overweight/obesity, sensitivity increased for both genders, mainly for male observers, but it remained lower than for female observers, although not statistically significant (76.6% for female observers vs. 66.7% for male observers, *p* = 0.146). Specificity and positive LR were higher for obesity, whereas negative LR was always lower for obesity/overweight.

**Table 3.** Sensitivity, specificity, and likelihood ratios of estimated obesity and overweight/obesity by observation, according to gender of observer, gender of participant, and age of participant (n = 254 observations).

| | Obesity | | | | Overweight/Obesity | | | |
|---|---|---|---|---|---|---|---|---|
| | | | Likelihood Ratio | | | | Likelihood Ratio | |
| | Sensitivity | Specificity | Positive (LR+) | Negative (LR−) | Sensitivity | Specificity | Positive (LR+) | Negative (LR−) |
| Gender of Observer | | | | | | | | |
| Female | 56.8% | 95.0% | 11.4 | 0.45 | 76.6% | 76.6% | 3.3 | 0.31 |
| Male | 14.3% | 98.7% | 11.0 | 0.87 | 66.7% | 81.5% | 3.6 | 0.41 |
| | *p* = 0.002 | *p* = 0.184 | | | *p* = 0.146 | *p* = 0.623 | | |
| Gender of Participant | | | | | | | | |
| Female | 50.0% | 92.9% | 7.0 | 0.54 | 65.6% | 100% | - | 0.34 |
| Male | 38.1% | 100% | - | 0.62 | 80.0% | 68.0% | 2.5 | 0.29 |
| | *p* = 0.411 | *p* = 0.014 | | | *p* = 0.029 | *p* = 0.002 | | |
| Age of Participant (years) | | | | | | | | |
| 18–34 | 20.0% | 100% | - | 0.80 | 79.2% | 91.7% | 9.5 | 0.23 |
| 35–54 | 37.5% | 97.6% | 15.6 | 0.64 | 66.2% | 78.1% | 3.0 | 0.43 |
| ≥55 | 54.2% | 93.4% | 8.2 | 0.49 | 76.8% | 61.1% | 2.0 | 0.38 |
| | *p* = 0.161 | *p* = 0.206 | | | *p* = 0.249 | *p* = 0.059 | | |
| <50 or ≥50 years | | | | | | | | |
| <50 | 33.3% | 100% | - | 0.67 | 69.2% | 85.4% | 4.7 | 0.36 |
| ≥50 | 50.0% | 93.0% | 7.1 | 0.54 | 75.5% | 65.4% | 2.2 | 0.37 |
| | *p* = 0.198 | *p* = 0.014 | | | *p* = 0.350 | *p* = 0.046 | | |

Regarding the gender of the participant, sensitivity to detect overweight/obesity was higher among male participants (80.0% vs. 65.6%, *p* = 0.029). Specificity for obesity was also higher among men (100% vs. 92.9%, *p* = 0.014), whereas specificity for overweight/obesity was higher among women (100% vs. 68%, *p* = 0.002). The LR− was similar among men and women, and lower for overweight/obesity (LR−women = 0.34; LR−men = 0.29) than for obesity (LR−women = 0.54; LR−men = 0.62).

Although not statistically significant, the older age of the participant increased sensitivity to detect obesity (from 20.0% for 18–34 years old to 54.2% for 55 years old or more). When age was regrouped into only two categories (<50 years and ≥50 years), the sensitivity to detect obesity and overweight/obesity together was similar to the sensitivity values when three age categories were considered. For obesity, the LR+ ranged between 7.1 and 15.6, and for overweight/obesity, it was highest for participants aged 18–34 years (LR+ = 9.5), ranging between 2.0 and 4.7 in the remaining groups.

### 3.4. Inter-Observer Reliability Analysis

As shown in Table 4, there was substantial agreement between observers for height ($\kappa$ = 0.63). For weight estimates ($\kappa$ = 0.46), BMI estimates ($\kappa$ = 0.52), and Stunkard Figures ($\kappa$ = 0.30), the judgement reliability was moderate to low. Still, agreement between observers was substantial when identifying subjects with or without overweight/obesity (BMI <25 or $\geq$25 kg/m$^2$, $\kappa$ = 0.67).

**Table 4.** Concordance between observers regarding estimates of height, weight, BMI, and Stunkard Figures (n = 127 paired observations).

|  | Kappa | 95%CI |
|---|---|---|
| **Height (cm)** | | |
| $\leq$154 155–164 $\geq$165 | 0.63 | 0.49–0.76 |
| **Weight (kg)** | | |
| $\leq$54 55–74 $\geq$75 | 0.46 | 0.31–0.62 |
| **BMI (kg/m$^2$)** | | |
| <18.5 18.5–24.9 25.0–29.9 $\geq$30 | 0.52 | 0.39–0.64 |
| <25 $\geq$25 | 0.67 | 0.50–0.83 |
| Stunkard Body Figures (1 to 9) | 0.30 | 0.22–0.38 |

BMI, body mass index.

## 4. Discussion

Our findings showed that visual estimation of obesity among adult individuals by trained observers using a body image scale was moderately sensitive (72.8% for overweight/obesity; 41.4% for obesity) and highly specific (78.4% for overweight/obesity; 96.4% for obesity). These results are similar to those reported in studies where healthcare providers estimated patients' weight with an accuracy of 40% to 70% [23–30].

When combining obesity status with overweight, in order to test the ability of observers to correctly distinguish normal weight from overweight (including obesity), sensitivity increased. It was more likely to correctly classify overweight overall (including obesity) than obesity alone, which may be due to the underestimation of obesity, as reported elsewhere [31–35]. Underestimation of obesity more than overweight might be explained by normal visual perceptual biases as contraction bias, which means that the weight of obese bodies will be increasingly underestimated as the BMI increases, and by Weber's law, which predicts that change in body size will become progressively harder to detect as their BMI increases [36–39]. These normal visual perceptual biases are supported by visual normalization theory, in which exposure to larger body sizes changes the range of body sizes that are perceptually judged as being "normal" [40–43]. We should also consider the effect of weight bias caused by negative beliefs about obese individuals and related stereotypes. Data indicate that a wide range of media portray overweight and obese individuals in a stigmatizing manner [44] and, additionally, even health professionals whose careers emphasize research or clinical management of obesity exhibit significant pro-thin and anti-fat bias, indicating a pervasive and powerful stigma [45].

When assigning weight-based descriptors to individuals to assess physician perception of patient weight, women physicians recognized the overweight status of their patients

more readily than men [46]. In our study, women were also more accurate in visual body weight estimation than men, as female observers estimated obesity with significantly higher sensitivity than male observers. However, the small number of observers does not allow us to draw bold conclusions about this.

The gender of the participant has been shown to be associated with differences in the estimation of overweight/obesity. Overweight/obese men were more accurately classified than overweight/obese women. This finding is similar to that reported elsewhere, where physicians of both genders were also less likely to recognize overweight status among female patients [46].

For this study, previous training for observers was performed. Even so, inter-observer reliability results indicate that there is room for improvement in training procedures (namely the inclusion of real people as part of the practical exercises), in order to maximize the concordance between observers, thus reaching more reliable estimations. Testing the concordance between a larger number of observers, as performed in a previous study where the accuracy of visual estimation of weight in geriatric patients by panels of two and three observers was compared [47], would also be an important recommendation for future work.

The main limitations of the study are the small sample size and the use of two-dimensional figures from the Stunkard scale, although validation studies have shown high correlations between self-reported BMI using this scale's body size figures and real BMI [18]. The small number of observers limited the conclusions in regard to the potential effect that the observers' characteristics may have on weight status assessment. However, to our knowledge, this is the first study that intends to classify the accuracy of body weight status assessment in adults by paired trained observers using body image scales. As such, the exploratory and innovative nature of the data presented should be highlighted.

The development, improvement, and validation of this new and easy-to-use methodology for structured observation may be very useful when anthropometric measurement is not possible. Although anthropometric measurement is the established gold standard for obtaining weight and height information, in some studies, this methodology can pose some challenges to achievable data collection, namely due to logistical constraints. Some examples include the following: (1) the study setting may not allow for the existence of the ideal privacy conditions for the individuals to be measured; (2) in some regions, cultural and/or religious barriers to body measurement may exist, which can lead to low participation rates; or (3) in some countries (especially low-income countries), the human and material resources necessary for anthropometric measurement may not be sufficient. Also, the incorporation of pictorial images with known BMI, as well as professionals trained in structured observation using such images, could address some of the limitations associated with self-reported measures, especially related to the underestimation of weight and BMI, and the consequent misidentification of overweight and obese subjects.

## 5. Conclusions

In conclusion, this study validates the estimation of weight status by direct observation. Observers were able to distinguish individuals of normal weight from overweight/obese individuals with high sensitivity and specificity, and substantial inter-observer reliability. Nevertheless, the identification of overweight/obesity status by structured observation by trained observers has high potential for future improvement through enhanced training techniques. This study represents an important step towards the future utilization of this innovative method for a more simple and fast collection of accurate weight status data.

**Author Contributions:** N.L. and P.P. designed the study. T.J. was responsible for conducting the study. I.d.C. and P.P. supervised the conduction of the study. T.J., N.L., S.S. and P.P. performed the analysis and interpretation of the results. T.J. and S.S. drafted the manuscript. All authors have read and agreed to the published version of the manuscript.

**Funding:** This study was financed through national funding from the Foundation for Science and Technology—FCT (Portuguese Ministry of Science, Technology and Higher Education), under the projects UIDB/04750/2020 and LA/P/0064/2020.

**Institutional Review Board Statement:** The study was conducted in accordance with the Declaration of Helsinki, and approved by the Ethics Committee of Hospital Center Lisbon-North/Faculty of Medicine of Lisbon (reference number 92/18; date of approval: 11/04/2018).

**Informed Consent Statement:** Informed consent was obtained from all subjects involved in the study. Participation was voluntary and anonymous. Participants were identified by a numeric code. Anthropometric measures were performed in closed specific zones in order to ensure privacy.

**Data Availability Statement:** The datasets used and/or analyzed during the current study are available from the corresponding author upon reasonable request.

**Acknowledgments:** The authors would like to acknowledge the work of the trained observers, without which this study would not have been possible. We also thank all participants of the study.

**Conflicts of Interest:** The authors declare no conflict of interest.

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
