# Peer review of "Accuracy of Assessing Weight Status in Adults by Structured Observation"

_applsci, doi:10.3390/app13148185_

Round 1
Reviewer 1 Report
T. Jorge et al present an evaluation of a method to estimate weight and height of more or less randomly selected participants of a study carried out in a public laboratory. Although they use the distribution independent chi squared test for comparison of groups, they show results of estimation as mean and standard deviation, which are clearly dependent on normal distributions. Therefore results should be presented as median and percentiles. Furthermore, although it would have been easy to measure weghts and height directly by scale and balance, the author do not present a clear (rank)correlation of estimations of trainers and results obtained by direct methods. Therefore the paper is acceptable after at least major revisions.
The English language of the paper is of acceptable quality. But as I am not a native speaker of English myself, I recommend revision by a professional language editor.
Reviewer 2 Report
This manuscript aimed to test the ability of trained observers to accurately estimate weight status in adults using structured observation.
The results are well presented, statistically supported and with discussion consistent with the results obtained.
Minor revisions:
The references used throughout the manuscript, and especially in the introduction, are very old (the most recent one is from 2017). It is necessary to include more recent references to demonstrate the topicality of the researched subject.
Reviewer 3 Report
The authors present a study that examines the efficacy of structured observation in accurately assessing weight status in adults, with a specific focus on the capability of trained observers. The findings illustrate that the trained observers displayed commendable sensitivity and specificity when distinguishing between normal weight and overweight/obesity, while also showcasing substantial interrater reliability. This study's innovative approach holds promise for further refinement through the implementation of advanced training techniques. Although this research is intriguing, the authors are advised to address several considerations for manuscript revision, as outlined below:
1. I suggest incorporating a statement about the significance of this study in the concluding remarks of the abstract to enhance its overall impact.
2. The reviewer acknowledges the author's intent but advises the author to emphasize the specific application scenarios of this study in the introduction to facilitate readers' better understanding of the purpose and significance of the research.
3. In general, when conducting research involving human data, authors are required to provide ethical approval. However, this study does not present the associated ethical documentation. Alternatively, the author may have specific reasons or explanations for this omission.
4. The author should provide an explanation regarding the criteria for selecting observers, including their qualifications and any other relevant professional requirements.
5. The author is encouraged to provide a more detailed explanation of the specific process involved in estimating measures through observation. It would be helpful for readers to understand if there are any reference points or control elements involved in this process, allowing them to grasp and apply the information more effectively.
6. I suggest reiterating the significance and potential applications of this study in the discussion and conclusion sections. Emphasizing these aspects will help underscore the importance and potential value of the research.
Round 2
Reviewer 1 Report
The revised manuscript is now worth publication.
Language quality is good enough that main objectives of the work are understandable.
Reviewer 3 Report
The authors have made appropriate revisions to the manuscript, rendering it suitable for acceptance and publication.